# Diffusion-limited association of disordered protein by non-native electrostatic interactions

Jae-Yeol Kim [1], Fanjie Meng[1], Janghyun Yoo[1] & Hoi Sung Chung [1]

Intrinsically disordered proteins (IDPs) usually fold during binding to target proteins. In contrast to interactions between folded proteins, this additional folding step makes the binding process more complex. Understanding the mechanism of coupled binding and folding of IDPs requires analysis of binding pathways that involve formation of the transient complex (TC). However, experimental characterization of TC is challenging because it only appears for a very brief period during binding. Here, we use single-molecule fluorescence spectroscopy to investigate the mechanism of diffusion-limited association of an IDP. A large enhancement of the association rate is observed due to the stabilization of TC by non-native electrostatic interactions. Moreover, photon-by-photon analysis reveals that the lifetime of TC for IDP binding is at least two orders of magnitude longer than that for binding of two folded proteins. This result suggests the long lifetime of TC is generally required for folding of IDPs during binding processes.

[1] Laboratory of Chemical Physics, National Institute of Diabetes and Digestive and Kidney Diseases, National Institutes of Health, Bethesda, MD 20892-0520, USA. Correspondence and requests for materials should be addressed to H.S.C. (email: chunghoi@niddk.nih.gov)

Intrinsically disordered proteins (IDPs) play important roles in various dynamic cellular processes such as gene transcription and signal transduction[1,2]. Since IDPs are unstructured and flexible, they can interact with multiple binding partners[1–4]. The kinetics of these interactions are, therefore, a key factor in the tight regulation of a complex binding network. The rates of association and dissociation of these proteins determine how fast a system can respond to external environmental changes[2,5,6]. The rate of a bi-molecular reaction in solution is limited by the relative diffusion of the two reacting molecules (i.e., the Smoluchowski limit) when the reaction occurs as soon as the molecules make contact (isotropic reactivity)[7]. In the Smoluchowski limit, the association rate coefficient ($k_A$) ranges from $10^9$ to $10^{10}$ $M^{-1} s^{-1}$. However, this fast association is not possible for binding of macromolecules because the fraction of reactive areas (i.e., binding interfaces) is much smaller. Due to the orientational constraint, $k_A$ can be smaller by several orders of magnitude, $10^5$–$10^6$ $M^{-1} s^{-1}$[8,9]. In IDP binding, the association is expected to be even slower because folding should occur during the binding process and an IDP may dissociate easily before folding even if it encounters a target protein. However, there are number of IDP binding systems that exhibit extremely fast association[10–14]. In this work, we investigated the molecular mechanism of such fast binding: association of the transactivation domain (TAD) of the tumor suppressor protein p53 and the nuclear coactivator binding domain (NCBD) of CREB-binding protein (CBP)[15] using single-molecule FRET spectroscopy (Fig. 1a). The transcription activity of p53 is controlled by binding of many proteins to the unfolded N-terminal TAD such as the E3 ubiquitin ligase MDM2 and various domains of the general transcriptional coactivator CBP including NCBD[16]. NCBD is also a flexible protein like a molten-globule[17–19] and interacts with multiple binding partners[20] including TAD and the activation domain of SRC-3 (ACTR)[21,22].

Unbound TAD is largely disordered, but three short helices are formed and the chain wraps around NCBD during binding (Fig. 1a). Unlike the binding of two folded proteins, therefore, binding of TAD involves large conformational changes, and it resembles a spontaneous protein folding process. Most importantly, binding pathways are expected to be highly diverse[23,24] as transition paths in protein folding, which can be probed only by watching individual molecules[25–27]. Characterization of this heterogeneity (i.e., distribution of binding pathways) is the key to understanding the detailed binding mechanism. We define a transient complex (TC, also known as encounter complex[28–31]) as a representative state appearing in the collection of binding pathways that lead unbound (disordered) TAD to a bound complex. We aim to describe the mechanism of diffusion-limited association of TAD and NCBD in terms of the properties of the TC. So far, the average picture of intermediate species during binding including encounter complexes has been characterized by NMR spectroscopy[32] such as relaxation dispersion[33] and paramagnetic relaxation enhancement[29,34]. The analysis of photon trajectories shows that the lifetime of TC of TAD and NCBD binding is much longer than that of the association of two folded proteins. The long lifetime results from the stabilization of TC by non-native electrostatic interactions, which makes diffusion-limited association possible because disordered TAD can rearrange and fold rather than dissociating quickly.

## Results

### Binding of TAD and NCBD depends strongly on ionic strength.
In single-molecule FRET experiments, we immobilized TAD, which was labeled with a donor (Alexa 488) and an acceptor (Alexa 647) on a glass surface, and incubated with unlabeled NCBD in solution (Fig. 1a) (see Supplementary Fig. 1 and Methods for the sequences of the protein constructs and details of protein expression and purification). The concentration of NCBD was varied to be near the equilibrium dissociation constant at each experimental condition so that the bound and unbound populations of TAD are comparable and both binding and dissociation transitions are frequently observed. Since TAD is disordered, the FRET efficiency of unbound TAD is low compared to that of the bound state, in which the end-to-end distance is much shorter (Fig. 1a).

IDPs are often highly charged, possibly to avoid aggregation[5], and binding partners are oppositely charged. Since the net charges of TAD and NCBD are −10 and +6, respectively, a strong electrostatic attraction is expected, which should facilitate the association. Therefore, we first investigated this effect by changing the ionic strength of the solution. Fig. 2a shows the binned fluorescence trajectories, FRET efficiency ($E$) trajectories, and FRET efficiency histograms measured at different NaCl concentrations (10 mM Tris, pH 7) (see Supplementary Fig. 2 for the FRET efficiency histograms obtained from freely diffusing molecules). At 0 mM NaCl, the FRET efficiency trajectory shows clear transitions between two levels, $E \sim 0.6$ and $E \sim 0.25$, which correspond to the bound and unbound states, respectively, indicating NCBD keeps binding and dissociating. As the NaCl concentration is increased, the bound and unbound states become less clear and the two peaks in the FRET efficiency histogram merge into a single peak. A narrow single peak at high ionic strength indicates that binding and dissociation are fast and $E$ is averaged by multiple transitions occurring during the bin time of 1 ms as observed for fast-folding, two-state proteins[35–37]. In order to obtain accurate parameters, we used a maximum likelihood method that extracts the FRET efficiencies of the bound and unbound states and apparent association and dissociation rates[38] (see Supplementary Fig. 3 and Methods for the kinetic models and the details of analysis methods). Since photon trajectories are analyzed directly without binning, it is possible to extract accurate parameters even for the case that no transition is identifiable in the binned trajectory at the highest NaCl concentration (150 mM, Fig. 2a). In addition, frequent acceptor photoblinking on the microsecond time scale can also be modeled in the analysis, which is necessary for the accurate determination of the parameters. The results are summarized in Fig. 2c and Supplementary Table 1. The FRET efficiency of the unbound state ($E_U$) gradually increases with the increasing NaCl concentration while that of the bound state ($E_B$) remains unchanged, indicating that disordered TAD becomes more compact because of reduced electrostatic repulsion at higher ionic strength[39]. Both dissociation ($k_D$) and apparent association rates ($k_{A,app}$) increase with the increasing NaCl concentration, consistent with the changes of the shape of the FRET efficiency histograms (Fig. 2a) and the increased relaxation rate (sum of the apparent association and dissociation rates) of the donor–acceptor cross-correlation function (Supplementary Fig. 4 and Supplementary Table 1). However, the association rate coefficient ($k_A$) of a bi-molecular reaction (the reaction is pseudo-first order because [TAD] « [NCBD]), which is obtained by dividing $k_{A,app}$ by the NCBD concentration, actually decreases as the NaCl concentration is increased (Fig. 2c).

### Diffusion-limited binding of TAD via formation of stable TC.
The strong ionic strength dependence of TAD–NCBD binding is expected from the large opposite net charges of the two proteins because the charge screening effect at high ionic strength will reduce $k_A$. However, it is unexpected that $k_A$ at 0 mM NaCl reaches $3.8 \times 10^9$ $M^{-1} s^{-1}$ (Fig. 2c), which is the regime of the

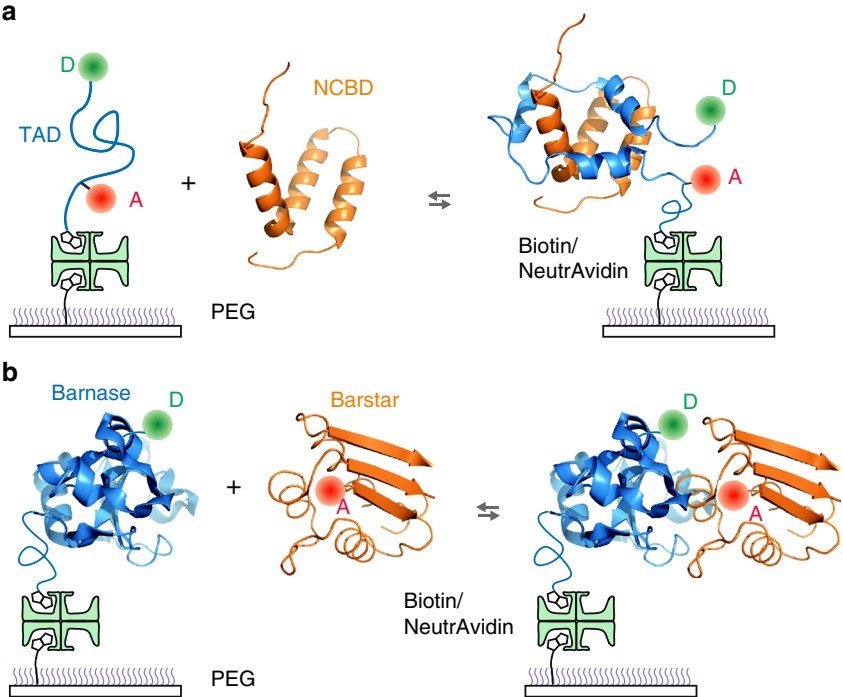

**Fig. 1** Binding experiment of immobilized proteins. **a** Alexa 488 (D, donor) and Alexa 647 (A, acceptor) are attached to cysteine at the C-terminus and 4-acetylphenylalanine at the N-terminus of TAD, respectively. Dye-labeled molecules were immobilized on a polyethylene glycol (PEG)-coated glass surface via a biotin–NeutrAvidin linkage and incubated with unlabeled NCBD in solution. **b** Donor-labeled barnase was immobilized on a glass surface and incubated with acceptor-labeled barstar

Smoluchowski limit. Despite the facilitation of binding by electrostatic attraction, this fast association is very unlikely unless a stable TC exists, in which two molecules are held together long enough for unfolded TAD to fold before separating from NCBD[40]. The lifetime of TC should be shorter than the bin time of 1 ms because the transitions appear instantaneous in Fig. 2a. For a better time resolution, we raised the laser illumination intensity by a factor of 5–10 and reduced the bin time to 100–200 μs (Fig. 3a)[35,41,42]. As indicated by yellow shades, there are several bins (200 μs bin time) with intermediate FRET efficiencies between the bound ($E \sim 0.6$) and unbound ($E \sim 0.25$) states, suggesting that an additional state exists between the bound and unbound states. Since this state with an intermediate FRET efficiency is not always obviously detectable in a binned trajectory, we performed a maximum likelihood analysis of photon trajectories with the three-state model including the bound state, TC, and the unbound state (six states including the acceptor dark state, see Supplementary Fig. 3 and Methods) as shown in Fig. 3b.

At 0, 10, 30, and 60 mM NaCl, there are peaks in the likelihood plot that are significantly higher than the 95% confidence level (Fig. 3c), and therefore, the lifetime of TC, $t_{TC}$ can be determined from the time at the maximum with high confidence[35]. On the other hand, there is no peak above this confidence level at 150 mM NaCl. In this case, only the upper bound of the lifetime can be determined[35]. At 90 mM, the peak is at the confidence level, and we determined both the time at the maximum and the upper bound for comparison (Supplementary Table 2). (See Methods and Supplementary Fig. 5 to find the measurable lifetimes given the amount of data and other parameters.) The analysis in Fig. 3c assumes that the FRET efficiency of TC ($E_{TC}$) is equal to the average of those of the bound and unbound states ($E_{TC} = (E_B + E_U)/2$). To check the validity of this assumption, we calculated the likelihood functions with various $E_{TC}$ values. Supplementary

Figure 6 shows that the likelihood is higher at $E_{TC}$ lower than the midway value, suggesting that TAD is largely unstructured in the TC (0 mM NaCl). Therefore, we also determined $t_{TC}$ and $E_{TC}$ simultaneously that maximize the likelihood (see Methods for the analysis details). $t_{TC}$ determined in this way is generally longer as summarized in Fig. 3e. $E_{TC}$ increases with the increasing NaCl concentration similar to $E_U$ (Fig. 3e, inset), indicating unstructured TAD in TC also becomes more compact at higher ionic strength. $t_{TC}$ in both analyses show a decreasing trend with increasing ionic strength (Fig. 3e). This result clearly indicates that TC is stabilized by electrostatic interactions, which makes diffusion-limited association possible.

In this measurement, we particularly note that the lifetime of TC, 183 μs (or 630 μs when determined with $E_{TC}$) at 0 mM NaCl is very long compared to transition path times (barrier crossing time) of protein folding, which has been measured so far to be between 2 and 12 μs depending on the presence of internal friction[35,43,44]. Given the analogy between the binding pathway and the transition path, we questioned if this long $t_{TC}$ could be a unique characteristic of coupled binding and folding of IDPs.

**Lifetime of TC of barnase and barstar is extremely short**. In order to answer the above question, we investigated the binding of barnase and barstar as a comparison (Fig. 1b). Binding of these two proteins is very similar to that of TAD and NCBD in that $k_A$ of the wildtype is close to that of the Smoluchowski limit and is very sensitive to the ionic strength of the solution[45,46]. The difference is that both barnase and barstar are folded regardless of binding. Since there is very little conformational change during binding, a donor was attached to barnase and an acceptor was attached to barstar to monitor binding. Fig. 2b shows representative fluorescence and FRET efficiency trajectories measured

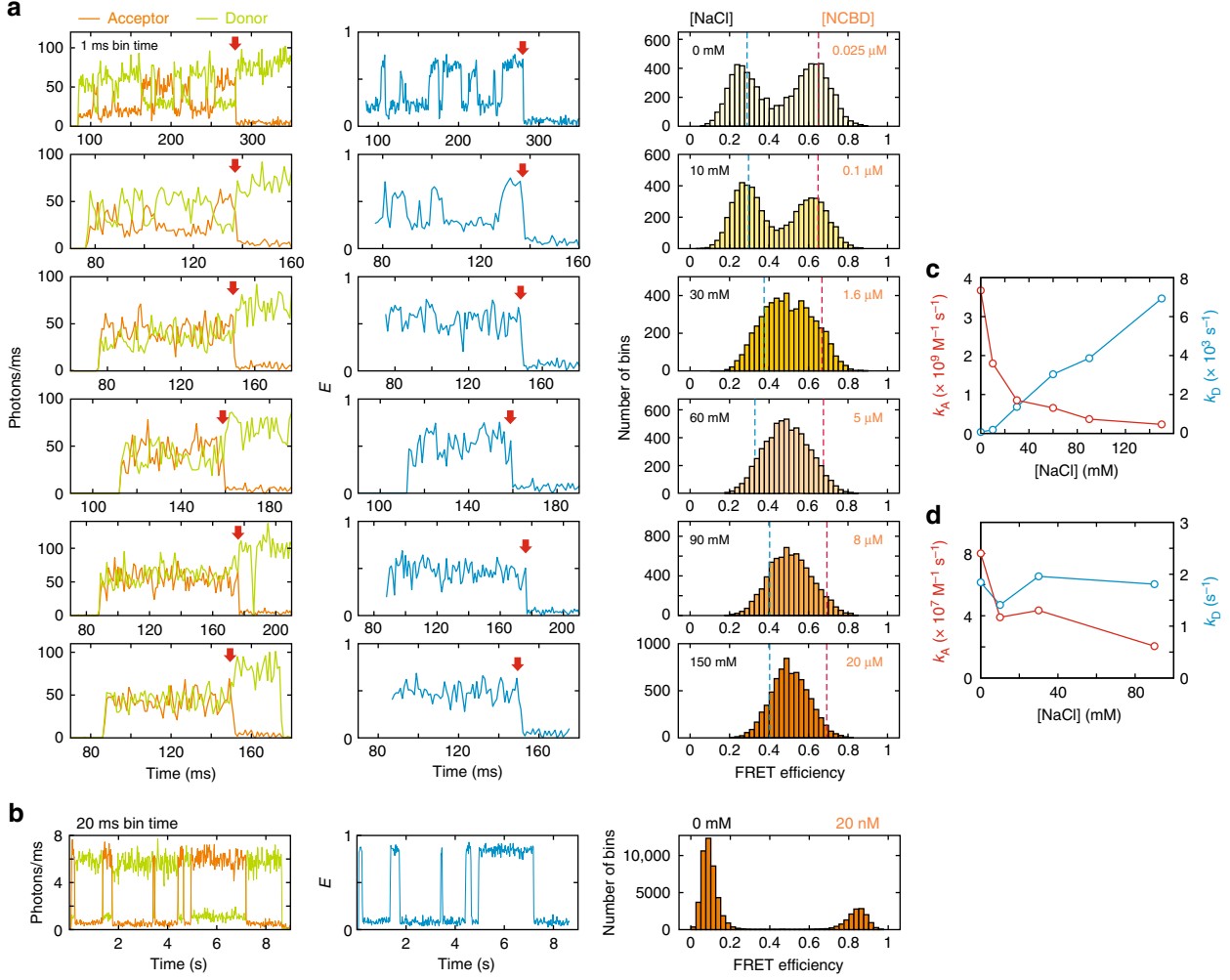

**Fig. 2** Ionic strength dependence of binding kinetics. **a** Donor and acceptor fluorescence trajectories (left), FRET efficiency trajectories (middle), and FRET efficiency histograms (right) of TAD/NCBD binding at various NaCl concentrations. Bin time is 1 ms and quoted numbers in the FRET efficiency histograms are NaCl (left) and NCBD (right) concentrations. Red arrows indicate photobleaching of the acceptor. **b** Donor and acceptor fluorescence trajectories (left), FRET efficiency trajectory (middle), and FRET efficiency histogram (right) of binding of barnase and barstar (20 nM) at 0 mM NaCl. Bin time is 20 ms. **c**, **d** Association (red) and dissociation (blue) rate coefficients of **c** TAD/NCBD and **d** barnase/barstar binding

at 0 mM NaCl, which exhibits multiple association and dissociation events. Since there is no resonance energy transfer before binding, the FRET efficiency of the unbound state is the same as that of the donor-only state. ($E$ is slightly higher than 0 because of the donor leak into the acceptor channel and direct excitation of the acceptor attached to barstar in solution.) In this case, acceptor photobleaching is not distinguishable from dissociation, and will reduce the apparent residence time in the bound state and result in the increased dissociation rate. Therefore, the two-state model parameters were determined using the maximum likelihood method with a correction for acceptor photobleaching (see Methods, Supplementary Fig. 7, and Supplementary Table 3 for the effect of acceptor photobleaching and its correction). $k_A$ at 0 mM NaCl is ~$10^8\,M^{-1}\,s^{-1}$, which is lowered by mutation and dye attachment compared to that of the wildtype. However, $k_A$ is very sensitive to the ionic strength while $k_D$ is unchanged (Fig. 2d), similar to the ionic strength dependence of the wildtype kinetics, suggesting that the binding mechanism is not altered. The most striking difference between the binding of barnase and barstar and binding of TAD and NCBD is found in the lifetime of TC (Fig. 3d). $t_{TC}$ of barnase and barstar binding is shorter than 2 μs (upper bound), which is

shorter than that of TAD and NCBD by two orders of magnitude (Fig. 3e).

**Non-native electrostatic interactions stabilize TC of TAD.** The above results, similar sensitivity of the association rate coefficient to the ionic strength but very different $t_{TC}$ of the two systems, can be explained by very different binding mechanisms. First, the binding interface of barnase and barstar comprises predominantly electrostatic interactions between oppositely charged sidechains (Fig. 4a). Vijayakumar et al.[47] have shown that increased $k_A$ compared to that of the basal rate (infinite ionic strength) results from the reduced free energy of the transition state (i.e., barrier height) due to the electrostatic interactions. In addition, the configurations of the two molecules at the transition state are very close to the structure of the bound state. The two molecules are shifted away slightly to accommodate a layer of water molecules and the relative orientation of the two binding interfaces can be only slightly tilted (within 3°). In other words, when two molecules approach with different orientations, binding would not happen (Fig. 4d). This interpretation is consistent with the recent MD simulation result by Plattner et al., which

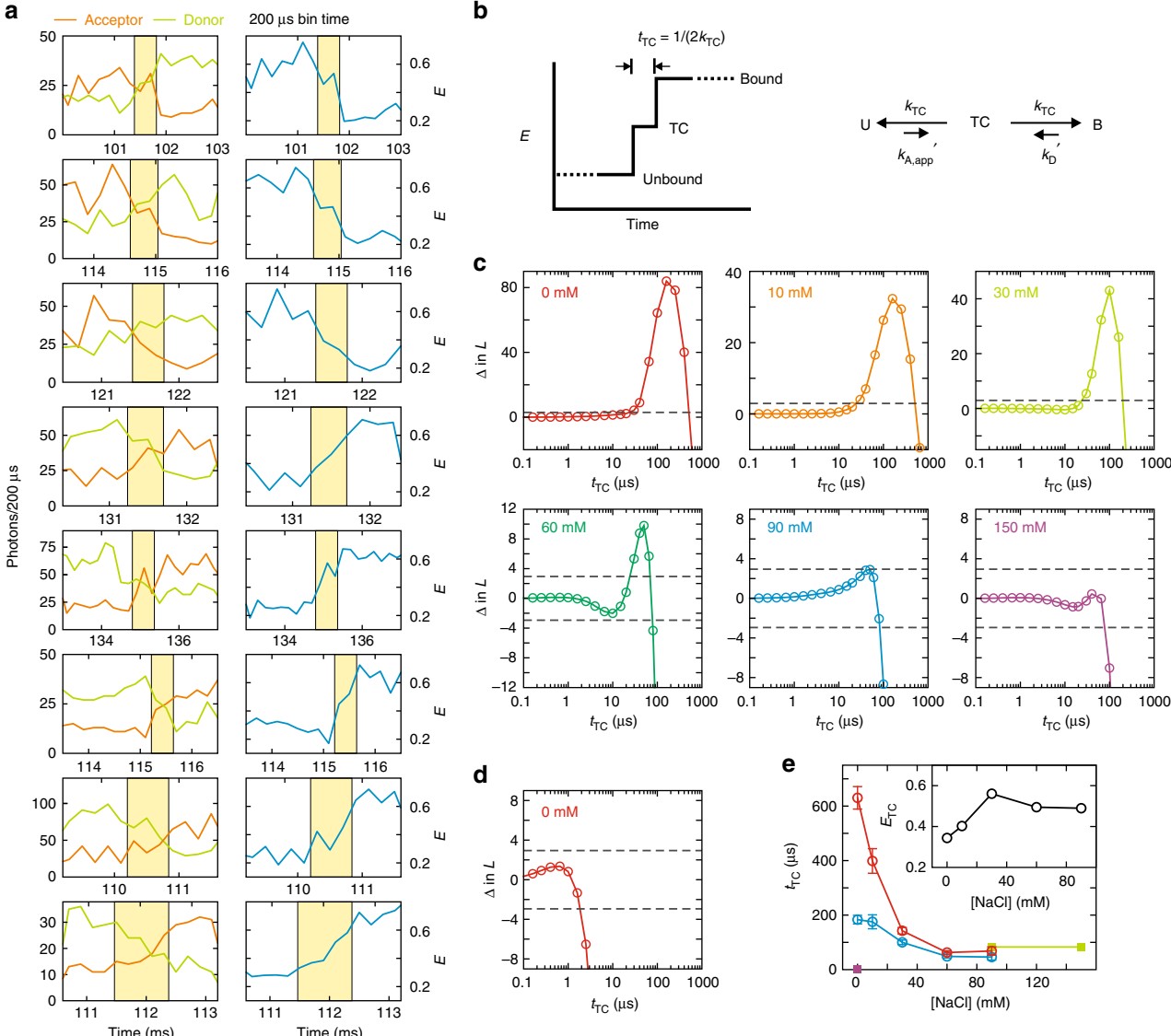

**Fig. 3** Measurement of the lifetime of transient complex. **a** Selected donor and acceptor fluorescence trajectories (left) and FRET efficiency trajectories (right) collected at high illumination intensity (200 μs bin time) that exhibit gradual changes in the fluorescence intensities and FRET efficiency (yellow shade) during association and dissociation. The accurate lifetime of TC was determined using the maximum likelihood analysis of photon trajectories without binning (see **e**). **b** Three-state model to determine the lifetime of TC ($t_{TC}$). The two rate coefficients of the transitions from TC to the bound and unbound states are set to be equal ($k_{TC}$) for the convenience of the analysis, which does not reflect the actual relative heights of the two barriers in Fig. 4c. **c** The difference of log-likelihood ($\Delta \ln L$) plots for binding of TAD and NCBD as a function of $t_{TC}$ at various NaCl concentrations (quoted numbers). $\Delta \ln L = \ln L(t_{TC}) - \ln L(0)$ compares the likelihood of the three-state model with a finite lifetime, $t_{TC}$, with the model with an instantaneous transition ($t_{TC} = 0$). The FRET efficiency of TC is assumed to be the average of the bound and unbound FRET efficiencies ($E_{TC} = (E_B + E_U)/2$). When the peak of the likelihood is significantly higher than the 95% confidence level ($\Delta \ln L = +3$, upper dashed line), $t_{TC}$ can be determined from the time at the maximum. When there is no significant peak, the upper bound of the lifetime can be determined from the time where $\Delta \ln L$ crosses the lower 95% confidence level ($\Delta \ln L = -3$, lower dashed line). **d** $\Delta \ln L$ plotted for binding of barnase and barstar at 0 mM NaCl. **e** The dependence of $t_{TC}$ on the NaCl concentration. $t_{TC}$ of TAD and NCBD binding is determined from either the maximum of $\Delta \ln L$ in **c** (blue) or by maximizing $\Delta \ln L$ with $E_{TC}$ as a free parameter (red, see Methods). Fitted values of $E_{TC}$ are shown in the inset. Green filled squares are the upper bound of $t_{TC}$ at 90 and 150 mM NaCl. A purple filled square is the upper bound of $t_{TC}$ of barnase and barstar binding at 0 mM NaCl in **d**. Fitting parameters are also listed in Supplementary Table 2. Errors are standard deviations obtained from the diagonal elements of the covariance matrix calculated at the maximum of the likelihood function

shows that the early intermediate states with different orientations would not affect overall binding kinetics significantly[48]. In the simulation, the lifetime of the transition state is 2 μs, which is close to the upper bound of the lifetime of TC. Although the lifetime of late intermediate states is ~10 μs, which is not detected as a significant peak in the likelihood plots with different $E_{TC}$ values in our analysis (Supplementary Fig. 8), the overall duration

of binding in the simulation is still much shorter than that of TAD and NCBD. The native-like transition state and other transient intermediate states explain why the lifetime of TC of barnase and barstar binding is very short compared to that of TAD and NCBD.

On the other hand, the binding interface of TAD and NCBD consists of hydrophobic interactions although both molecules are

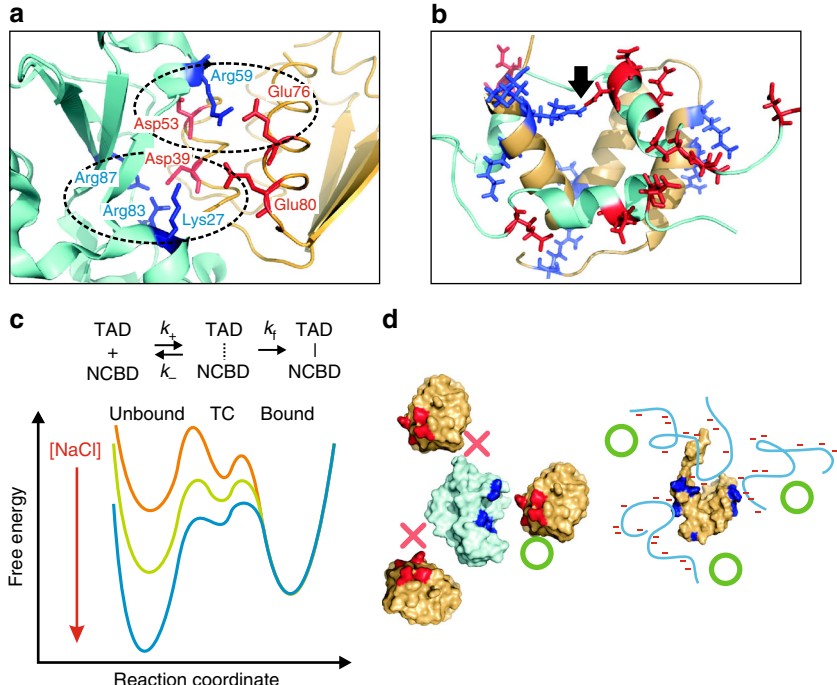

**Fig. 4** Binding mechanisms. **a** Electrostatic interactions in the binding interface of barnase and barstar. Blue and red sticks indicate positively charged side chains of barnase (cyan) and negatively charged side chains of barstar (orange), respectively. Dashed ellipses show two clusters of interacting residues. **b** Positively charged side chains (blue sticks) of NCBD (orange molecule) and negatively charged side chains (red sticks) of TAD (cyan molecule). Charged side chains are exposed to the solvent and do not interact except those in one salt bridge indicated by a black arrow. Binding interface is predominantly hydrophobic. **c** Kinetic scheme of the association via the formation of TC, and proposed free energy surface of TAD/NCBD binding with an arbitrary NCBD concentration at different NaCl concentrations. $k_+$ and $k_-$ are the association and dissociation rate coefficients of the formation of TC. $k_f$ is the folding rate of unfolded TAD in TC. $k_A = k_+ k_f/(k_f + k_-)$. As [NaCl] is increased, the formation of TC becomes slower (reduced $k_+$) due to the increased charge screening effect, and TC becomes less stable and dissociates more easily (increased $k_-$, lowered dissociation barrier) before TAD folds to form a fully bound complex. Both effects decrease $k_A$. **d** The transition state configurations (indicated by green O) of barnase (cyan) and barstar (orange) are very similar to that of the bound complex, and therefore, the configurations with unfavorable orientations will not result in association (indicated by red x). Positively and negatively charged side chains in the binding interface are colored in blue and red. On the other hand, the association can be initiated from different parts of NCBD (orange) via non-native interactions between charged residues (i.e., parallel binding pathways[23,24], indicated by green O). Cyan coils represent disordered TAD. Red bars indicate negative charges in TAD and positively charged side chains of NCBD are colored in blue

highly charged[15]. All charged side chains except two participating in the native salt bridge (indicated by a black arrow in Fig. 4b) are located on the opposite side of the binding interface (Fig. 4b). In this case, the formation of the native interactions will be hardly affected by the ionic strength of the solution. Therefore, the electrostatic interactions must be involved in the formation of the TC and these interactions are non-native, although non-native hydrophobic interactions can also contribute[33,49]. The long lifetime of TC indicates that once non-native contacts are formed between charged side chains, two molecules would stick together so that TAD rearranges and folds on the surface of NCBD without dissociation (small $k_-$ in Fig. 4c). In this way, the association can be diffusion-limited. As the ionic strength is increased, the formation of TC will be slower (reduced $k_+$ in Fig. 4c) due to the increased charge screening effect, and TC becomes less stable and dissociates more easily before TAD folds to form a fully bound complex (increased $k_-$). Both effects will decrease the association rate ($k_A = k_+ k_f/[k_f + k_-]$) (Fig. 4c). Non-native interactions at low ionic strength also suggest slow diffusion along the reaction coordinate, which contributes to the increased $t_{TC}$ along with the stability of TC similar to the increased folding transition path time of a designed protein, $\alpha_3 D$ by non-native salt-bridge formation[44].

Since the unbound TAD is very extended due to the electrostatic repulsion as indicated by the low FRET efficiency given the short chain length of TAD in this study (49 residues), the fly-casting type enhancement of the association rate is expected[50]. This enhancement will also be reduced by increased ionic strength, which makes disordered TAD more compact[39].

## Discussion

As stated in the Introduction, the most crucial information for understanding binding mechanisms of IDPs is contained in binding pathways including the formation of TC, in which conformational changes actually occur, analogous to the transition path in protein folding[51]. However, diversity in binding pathways and the short lifetime of TC make characterization very difficult. As a first step toward this direction, we employed photon-by-photon analysis for single-molecule FRET with high time resolution to systematically measure the lifetime of TC of TAD and NCBD as a function of ionic strength and describe the binding mechanism. We found that both binding kinetics and lifetime of TC are very sensitive to the ionic strength of the solution (Figs. 2, 3). More importantly at low ionic strength, the association is diffusion-limited, which is very unlikely to happen in IDP binding, while the lifetime of TC is unexpectedly long, i.e., several hundred microseconds. Our interpretation is that diffusion-limited association of TAD requires a long lifetime of TC, during which unstructured TAD in TC can fold without dissociation. In contrast, the lifetime of TC of barnase and barstar is at least two

orders of magnitude shorter because there is no folding step during binding (Fig. 3). Moreover, we found that TC of TAD and NCBD is stabilized by non-native electrostatic interactions, which is also required for fast association.

Sugase et al. have found that the association rate can be enhanced by non-native hydrophobic interactions for a much slower binding system (binding of the disordered phosphorylated kinase inducible activation domain (pKID) and the KIX domain of CBP) using NMR relaxation dispersion experiments[33]. $\phi$-Value analysis has also shown the involvement of non-native interactions for the same system[49]. The enhancement of the association by non-native interactions have also been found for various systems in simulations[23,24,52]. Therefore, the formation of a stable TC via non-native interactions may be a general mechanism for IDP binding. Although non-native hydrophobic interactions can contribute[33,49], we would like to point out that non-native electrostatic interactions should prevail since IDPs usually possess a large number of charged residues[53] while their binding interfaces are hydrophobic[54] (e.g., amphipathic helix formation).

It is also worth noting that the enhancement of association by non-native interactions is an important mechanism for IDP binding, whereas in protein folding, non-native interactions do not affect the folding mechanism[55], but slow the folding process[44,56]. In the case of diffusion-limited association at 0 mM NaCl, the lifetime of TC (several hundred microseconds) corresponds to the folding time of TAD ($t_{TC} = 1/[k_f + k_-]$ and $k_- \ll k_f$ in Fig. 4c). This time is actually much longer than folding times of many single domain ($\alpha$-helical) fast-folding proteins that can fold in several microseconds or even shorter[57]. The slow folding rate of TAD may result from the increased internal friction by non-native interactions as found for folding of a designed protein, $\alpha_3$D[44]. Between the two opposite effects of non-native interactions on the overall rate of diffusion-limited association ($k_A = k_+ k_f/[k_f + k_-]$), the enhancement by avoiding dissociation (reduced $k_-$)[23] is much larger than the reduction due to slower folding by a few fold (reduced $k_f$)[44].

In contrast to binding of barnase and barstar with a native-like TC, binding pathways are expected to be more heterogeneous because binding of TAD can be initiated from various parts of NCBD by non-native interactions (Fig. 4d) as observed in simulations for other systems[23,24]. For a more fundamental understanding of the binding mechanism, it will be very important to further investigate the characteristics of the TC including the heterogeneity of binding pathways for various IDP systems.

## Methods

**Protein expression, purification, and dye-labeling**. TAD was engineered as a fusion protein, 6His-GB1-Thb-Avi-UA-TAD-Cys. The sequence of 6His-GB1-Avi-UA-TAD-Cys consists of MGSSHHHHHHSSGMQYKLILNGKTLKGETTTEAV-DAATAEKVFKQYANDNGVDGEWTYDDATKTFTVTE (6His-56 residue long immunoglobulin-binding domain B1 of streptococcal protein G (GB1)), SSGLVPRGSGH MGMS (thrombin cleavage site flanked by spacer residues), GLNDIFEAQKIEWHE (biotin acceptor peptide termed Avi, Avidity LLC), SSGLVAGGGGSGGGGSGGGGS (long spacer) and UPLSQETFSDLWKLL-PENNVLSPLPSQ AMDDLMLSPDDIEQWFTEDPGPDC (UA-TAD-Cys). UA-TAD-Cys denotes the TAD of the tumor suppressor protein p53 (residue 13–61)[15] with the incorporation of an unnatural amino acid (UA, 4-acetylphenylalanine, SC-35005, SynChem Inc.) and a cysteine residue at the N- and C-terminus, respectively. NCBD (residue 2059–2117)[15,21], barnase, and barstar[58] were also designed as fusion proteins, 6His-GB1-Thb-NCBD, 6His-GB1-Thb-Avi-barnase (G34C/H102A), and 6His-GB1-Thb-Cys-barstar. The protein sequences are listed in Supplementary Fig. 1. All genes were cloned in pJ414 vector flanked by Nde1 and BamH1 restriction sites (DNA 2.0 Inc.).

For the expression of TAD with biotinylation and incorporation of UA, three plasmids, pJ414-TAD, pJ411-BirA, and pEvol were cotransformed into *Escherichia coli*. BL21(DE3) (200131, Agilent Technologies). pJ411-BirA encodes biotin ligase for biotinylation and pEvol encodes the amber codon (TAG) suppressor tRNA$_{CUA}$ and amino acetyl-tRNA synthetase for 4-acetylphenylalanine[59]. The transformed cells were grown in Luria-Bertani medium. Expression was induced at an absorbance of 0.7 monitored at 600 nm by adding final concentration of 1 mM

isopropylthiogalactoside (IPTG) for pJ414-TAD and pJ411-BirA, and 1 mM arabinose for pEvol in the presence of 1 mM 4-acetylphenylalanine and 50 µM d-biotin (B4501, Sigma-Aldrich). After further incubation at 37 °C for 4 h, cells were collected by centrifugation at 8000g for 10 min and the cell pellet was re-suspended in appropriate lysis buffers: 20 mM Tris–HCl pH 7.5, 8 M Urea, 10 mM DTT for TAD and NCBD, and 20 mM Tris–HCl pH 7.5, 400 mM NaCl, 10 mM DTT for barnase and barstar. Cells were lysed by brief sonication and resting cycles and the lysates were centrifuged at 50,000g for 40 min to collect supernatants.

From the supernatant, 6His-GB1-Thb-Avi-UA-TAD-Cys was purified by Ni-NTA affinity chromatography. Streptavidin Mutein column (03708152001, Roche Diagnostics) was also used to purify biotinylated proteins. The truncated protein without incorporation of UA (6His-GB1-Avi) was separated by size-exclusion chromatography using AKTA pure FPLC system with Superdex$^{TM}$75 10/300GL (GE Healthcare, Chicago, IL). The incorporation rate of UA was about 30%. Then, 6His-GB1 was cut by thrombin in 20 mM Tris–HCl pH 7.4, 100 mM NaCl, and removed using IgG Sepharose 6 Fast Flow column (GE Healthcare, Chicago, IL), which binds GB1 specifically. Avi-UA-TAD-Cys was recovered from the unbound flow-through. NCBD, Avi-barnase, and Cys-barstar were purified similarly by performing Ni-NTA purification and size exclusion chromatography followed by thrombin cleavage and removal of 6His-GB1.

Avi-UA-TAD-Cys was labeled with Alexa Fluor 488 maleimide (Alexa 488, A10254, Thermo Fisher Scientific) at the cysteine residue and Alexa Fluor 647 hydroxylamine (Alexa 647, A30632, Thermo Fisher Scientific) at UA. First, 100 µL of 100 µM Avi-UA-TAD-Cys in 50 mM Tris–HCl, pH 7.0, 6 M GdmCl was incubated with 1.5 mM TCEP for 2 h to reduce disulfide bonds, and 100 µg of Alexa 488 dissolved in 5 µL of DMSO was added. The reaction solution was incubated at room temperature for 4 h. Unreacted Alexa 488 dyes were removed by size-exclusion chromatography using superdex$^{TM}$Peptide 10/300GL. Alexa 647 was attached to UA using the formation of oxime between ketone and hydroxylamine[60]. The Alexa 488 labeled Avi-UA-TAD-Cys was incubated with Alexa 647 in an oxime reaction buffer, 50 mM sodium acetate, pH 4.0, 100 mM NaCl, 4 M GdmCl for 20 h at room temperature because oxime reaction requires low pH and shows slow reaction rate compared to the cysteine–maleimide reaction[60]. The reaction was quenched by raising pH to 7 and the unreacted dyes were removed using size-exclusion chromatography. Barnase and barstar were labeled with Alexa 488 maleimide and Alexa 647 maleimide, respectively, and purified similarly.

**Single-molecule spectroscopy**. Single-molecule FRET experiments were performed using a confocal microscope system (MicroTime200, Picoquant) with a 75 µm diameter pinhole, a beamsplitter (ZT405/488/635rpc, Chroma Technology), and an oil-immersion objective (UPLSAPO, NA 1.4, ×100, Olympus)[61]. Alexa 488 was excited by a 485 nm diode laser (LDH-D-C-485, PicoQuant) in the CW mode. Alexa 488 and Alexa 647 fluorescence was split into two channels using a beamsplitter (585DCXR, Chroma Technology) and focused through optical filters (ET525/50m for Alexa 488 and E600LP for Alexa 647, Chroma Technology) onto photon-counting avalanche photodiodes (SPCM-AQR-16, PerkinElmer Optoelectronics).

In the immobilization experiment, biotinylated TAD or barnase molecules were immobilized on a biotin-embedded, polyethylene glycol-coated glass coverslip (Bio_01, Microsurfaces Inc.) via a biotin (surface)-NeutrAvidin-biotin (protein) linkage[51]. The surface was initially incubated with NeutrAvidin (30 µg/mL) for 5 min and subsequently with the solution of TAD or barnase (80 pM) for 3 min. Then, immobilized molecules were incubated with unlabeled NCBD or barstar for binding. Molecules were illuminated at 3 and 0.3 µW for the measurement of the binding kinetics of TAD/NCBD and barnase/barstar systems, respectively. In the measurement of the lifetime of the TC, molecules were illuminated at 15–60 µW. All experiments were performed in 10 mM Tris buffer (pH 7) with 0–150 mM of NaCl. To reduce dye photobleaching and blinking, 2 mM cyclooctatetraene (COT), 2 mM 4-nitrobenzyl alcohol (NBA), 2 mM trolox[62,63], 10 mM cysteamine, and 100 mM β-mercaptoethanol[64] were used in addition to an oxygen scavenging system, 50 nM protocatechuate 3,4-dioxygenase (PCD, P8279-25UN, Sigma) and 2.5 mM 3,4-dihydroxybenzoic acid (PCA, 37580–25G-F, Sigma)[65].

In the free diffusion experiment, 80 pM of TAD was mixed with NCBD at various NaCl concentrations. The same chemicals used for the immobilization experiment were added to reduce dye photobleaching and blinking. To prevent protein sticking to a glass surface, 0.01% Tween 20 was used[66]. The solution was illuminated at 22 µW and fluorescence bursts were measured 10 µm above the glass surface.

All experiments were performed at room temperature (22 °C).

**Maximum likelihood analysis**. To determine the parameters for the two-state binding kinetics, we used the maximum likelihood method developed by Gopich and Szabo that analyzes photon trajectories directly without binning[38]. The likelihood function for the $j$th photon trajectory with records of photon colors and arrival times is

$$L_j = \mathbf{1}^T \prod_{i=2}^{N_j} [\mathbf{F}(c_i) \exp(\mathbf{K}\tau_i)] \mathbf{F}(c_1)\mathbf{p}_{eq} \qquad (\ ,1)$$

where $N_j$ is the number of photons in the $j$th trajectory, $c_i$ is the color of the $i$th photon (donor or acceptor), and $\tau_i$ is a time interval between the $(i − 1)$th and $i$th photons. $\mathbf{K}$ is the rate matrix, the photon color matrix $\mathbf{F}$ depends on color $c$ of a photon as $\mathbf{F}(\text{acceptor}) = \mathbf{E}$ and $\mathbf{F}(\text{donor}) = \mathbf{I} − \mathbf{E}$, where $\mathbf{E}$ is a diagonal matrix with the uncorrected (apparent) FRET efficiencies of the individual states on the diagonal, $\mathbf{1}^\text{T}$ is the unit row vector (T means transpose), and $\mathbf{p}_{eq}$ is the vector of equilibrium populations. The likelihood function was calculated by the diagonalization of $\mathbf{K}$ as described in ref. [38]. Practically, the total log-likelihood function of all trajectories was calculated by summing individual log-likelihood functions as $\ln L = \Sigma_j \ln L_j$.

For the two-state model (Supplementary Fig. 3a), there are four fitting parameters: the apparent FRET efficiencies of the bound and unbound states, $E_B$ and $E_U$, the apparent association rate, $k_{A,app}$, and dissociation rate coefficient, $k_D$. The association rate coefficient is $k_A = k_{A,app}/[\text{Protein}]$, where [Protein] is the concentration of the binding partner, NCBD or barstar. The matrix of FRET efficiencies, the rate matrix, and the vector of the equilibrium populations are given by

$$\mathbf{E} = \begin{pmatrix} E_B & 0 \\ 0 & E_U \end{pmatrix}, \mathbf{K} = \begin{pmatrix} -k_D & k_{A,app} \\ k_D & -k_{A,app} \end{pmatrix}, \mathbf{p}_{eq} = \begin{pmatrix} p_B \\ 1 - p_B \end{pmatrix} \quad (2)$$

$p_B = k_{A,app}/(k_{A,app} + k_D)$ is the equilibrium population of the bound state. At the illumination intensity to collect photon trajectories with a count rate of 50–100 ms$^{-1}$ for the determination of millisecond kinetics, there is frequent blinking. Donor blinking does not affect the result, but acceptor blinking increases the rate coefficients[37]. Therefore, we incorporated acceptor blinking in the model and used this four-state model (Supplementary Fig. 3b) to determine the parameters of binding of TAD and NCBD. In the four-state model, the matrices are given by[36]

$$\mathbf{E} = \begin{pmatrix} E_B & 0 & 0 & 0 \\ 0 & E_U & 0 & 0 \\ 0 & 0 & E_d & 0 \\ 0 & 0 & 0 & E_d \end{pmatrix}, \mathbf{p}_{eq} = \begin{pmatrix} p_B p_b \\ (1-p_B)p_b \\ p_B(1-p_b) \\ (1-p_B)(1-p_b) \end{pmatrix}$$

$$\mathbf{K} = \begin{pmatrix} -k_D - k_d & k_{A,app} & k_b & 0 \\ k_D & -k_{A,app} - k_d & 0 & k_b \\ k_d & 0 & -k_D - k_b & k_{A,app} \\ 0 & k_d & k_D & -k_{A,app} - k_b \end{pmatrix} \quad (3)$$

where $E_d$ is the FRET efficiency of the acceptor dark state, and $k_b$ ($k_d$) is the rate coefficient for the transition from the dark (bright) state to the bright (dark) state of the acceptor. $k_b$ is independent of the photon count rate. On the other hand, as the probability of the transition from the bright state to the dark state will increase linearly with the time spent in the excited state, $k_d$ is proportional to the photon count rate of each trajectory as $k_d = k_0\ (n/n_0)$, where $n$ is the average photon count rate of a photon trajectory and $k_0$ is the rate coefficient at the reference photon count rate, $n_0 = 100$ ms$^{-1}$. $p_b = k_b/(k_b + k_d)$ is the equilibrium population of the acceptor bright state.

For the binding kinetics of barnase and barstar, the effect of microsecond time scale acceptor blinking is negligible because the photon count rate is much lower and the binding kinetics is much slower[37]. Therefore, we used the two-state model (Eq. (2)). However, because the FRET efficiency of the unbound state is the same as that of the donor-only state (only the donor is attached to immobilized barnase), acceptor photobleaching is indistinguishable from dissociation. In this case, the waiting time in the bound state will be underestimated, which will increase the apparent dissociation rate and decrease the bound fraction, $p_B$. Therefore, we first determined $p_B$ from the fraction of the trajectories that begin with the bound state (high FRET efficiency), and then determined other two-state parameters by fixing $p_B$ (Eq. (2)). The accuracy of the parameters determined with and without fixing the bound fraction is compared in Supplementary Fig. 7. For the trajectories recolored with experimental parameters. As expected, $p_B$ and $k_{A,app}$ decrease and $k_D$ increases as the acceptor bleaching rate, $k_{bleach}$ increases when the parameters are extracted with $p_B$ as a free fitting parameter, whereas extracted rates are very accurate regardless of $k_{bleach}$ when $p_B$ is fixed.

In the measurement of the lifetime of the TC, $t_{TC}$, molecules were illuminated at higher laser intensity to collect photons at a count rate of 300–900 ms$^{-1}$. This high photon count rate made it possible to reduce the bin time to 100–200 μs, with which a brief residence of several hundred microseconds in TC of TAD and NCBD can be visualized for selected trajectories (Fig. 3a). A short segment of photon trajectories near each transition (400 μs to 3 ms) was then analyzed using the likelihood method. The difference of log-likelihood, $\Delta \ln L = \ln L(t_{TC}) − \ln L(0)$ compares the likelihood of the three-state model consisting of the bound state (B), TC, and unbound state (U) with a finite lifetime, $t_{TC}$, with that of the model with an instantaneous transition ($t_{TC} = 0$, i.e., two-state model) (see Supplementary Fig. 3 and the likelihood function in Eq. (4)). $\Delta \ln L = +3$ and $−3$ were used for the 95% confidence levels for the determination of $t_{TC}$ or an upper bound of the lifetime, respectively[35].

The likelihood function of the three- and six-state models in Supplementary Fig. 3c and d for the $j$th photon trajectory with a single transition is[35]

$$L_j = \mathbf{v}_{\text{fin}}^\text{T} \prod_{i=2}^{N_j} [\mathbf{F}(c_i) \exp(\mathbf{K}\tau_i)] \, \mathbf{F}(c_1) \mathbf{v}_{\text{ini}} \quad (4)$$

where $\mathbf{v}_{\text{ini}}$ and $\mathbf{v}_{\text{fin}}$ are the state vectors at the beginning and end of the trajectory, respectively. For the binding transition, for example, $\mathbf{v}_{\text{ini}} = [0\ 0\ 1]^\text{T}$ and $\mathbf{v}_{\text{fin}} = [1\ 0\ 0]^\text{T}$ for the three-state model (Supplementary Fig. 3c) and $\mathbf{v}_{\text{ini}} = [0\ 0\ p_b\ 0\ 0\ (1 − p_b)]^\text{T}$ and $\mathbf{v}_{\text{fin}} = [1\ 0\ 0\ 1\ 0\ 0]^\text{T}$ for the six-state model (Supplementary Fig. 3d). In the calculation of the likelihood function in Eq. (4), we reduced the apparent association and dissociation rates by a factor of 1000 to effectively eliminate the contribution from multiple transitions that are not resolvable, i.e., $k_D{}'\ (=k_D/1000)$ and $k_{A,app}{}'\ (=k_{A,app}/1000)$. This treatment is valid since we use the difference of the log-likelihood values ($\Delta \ln L$).

For the analysis of barnase/barstar binding, we used the three-state model (Supplementary Fig. 3c). The rate matrix is

$$\mathbf{K} = \begin{pmatrix} -k'_D & k_{TC} & 0 \\ k'_D & -2k_{TC} & k'_{A,app} \\ 0 & k_{TC} & -k'_{A,app} \end{pmatrix} \quad (5)$$

In this model, $t_{TC} = 1/2k_{TC}$. $E_B$ and $E_U$ were calculated for each transition and the likelihood function was calculated with $E_{TC} = (E_B + E_U)/2$ as a function of $t_{TC}$ as described in ref. [35] (Fig. 3d). $E_{TC}$ can be varied and the analysis with different $E_{TC}$ values is shown in Supplementary Fig. 8. We analyzed only the first binding event in each molecular trajectory because it is not possible to distinguish dissociation from acceptor photobleaching or photoblinking on the millisecond time scale. It is also impossible to distinguish between rebinding and transition from the acceptor dark state to the bright state.

For the determination of $t_{TC}$ of TAD and NCBD binding, we used the six-state model (Supplementary Fig. 3d) that includes acceptor dark states because acceptor blinking on the microsecond time scale prevents the accurate determination of $t_{TC}$ or the upper bound. In this model, each of the bound, unbound, and TC exists in both bright (fluorescing) and dark (non-fluorescing) states. All rate coefficients between three states in the acceptor dark state are assumed to be the same as those in the bright state. For this model, the rate matrix is given by[44]

$$\mathbf{K} = \begin{pmatrix} -k'_D - k_d & k_{TC} & 0 & k_b & 0 & 0 \\ k'_D & -2k_{TC} - k_d & k'_{A,app} & 0 & k_b & 0 \\ 0 & k_{TC} & -k'_{A,app} - k_d & 0 & 0 & k_b \\ k_d & 0 & 0 & -k'_D - k_b & k_{TC} & 0 \\ 0 & k_d & 0 & k'_D & -2k_{TC} - k_b & k'_{A,app} \\ 0 & 0 & k_d & 0 & k_{TC} & -k'_{A,app} - k_b \end{pmatrix}$$
$$(6)$$

In the analysis of TAD/NCBD binding, first, the bound and unbound segments were assigned using the Viterbi algorithm[67,68] for the two-state model, adapted for photon trajectories[69]. Since transitions with short residence times caused by acceptor blinking were frequently assigned due to the relatively low $E_U$ value, transitions with residence times longer than ~200 μs in both bound and unbound states were analyzed. When there are multiple transitions in a trajectory, each pair of segments with a single binding or dissociation transition was analyzed separately.

In the six-state analysis, before calculating $\Delta \ln L(t_{TC})$ as a function of $t_{TC}$, the blinking parameters $k_b$ and $p_b$, and the average FRET efficiencies of the bound and unbound states ($E_B$ and $E_U$) were first determined by maximizing $\Delta \ln L(0)$ (instantaneous transition model with acceptor blinking ($k_{TC} \rightarrow \infty$)). With these parameters, $\Delta \ln L(t_{TC})$ was calculated with fixed $E_{TC} = (E_B + E_U)/2$ as shown in Fig. 3c (upper block in Supplementary Table 2)[44]. In the analysis to determine $E_{TC}$ and $t_{TC}$ together (Fig. 3e, lower block of Supplementary Table 2), all six parameters ($E_B$, $E_U$, $k_b$, $p_b$, $E_{TC}$, and $t_{TC}$) were determined simultaneously by maximizing $\ln L$ ($t_{TC}$).

**Simulation of photon trajectories.** To validate the experimental measurement of the lifetime of TC, photon trajectories were simulated, re-analyzed, and the results of this analysis were compared with the experimental results (Supplementary Fig. 5). In this simulation, instead of generating completely new photon trajectories, the intervals between photons of the experimental photon trajectories are retained, and only the photon colors are erased and recolored. By generating photon trajectories in this way, it is possible to produce a simulated dataset that is most similar to the experimental data in terms of both the average and distribution of the detected photons per unit time, and the length of the trajectories.

In the data collected at high illumination intensity for the measurement of $t_{TC}$, trajectory segments that contained a single transition were recolored after dividing each segment into three regions: bound, TC, and unbound states. First, the region corresponding to TC was centered at the most probable instantaneous transition

photon interval found by the Viterbi algorithm (two-state model) in the analysis. The width of the TC region was randomly chosen (exponential distribution) with the mean value that is the same as the assumed lifetime of TC in the simulation. The two sides of the TC region were then assigned as bound and unbound states according to the states in the experimental trajectory, and the photons of each region were recolored using the FRET efficiencies of the three states: $E_B$, $E_{TC}$, and $E_U$. The probabilities for observing a donor and an acceptor photon are $(1 - E)$ and $E$, respectively. Finally, each simulated trajectory was divided into acceptor bright and dark state regions using the acceptor blinking kinetics parameters, and photons in the dark state were recolored using $E = 0.06$ (fraction of donor photons leaking into the acceptor channel).

**Correlation analysis**. The donor–acceptor cross-correlation function was calculated as

$$C_{DA}(\tau) = \frac{\overline{\langle N_D(t + \tau) N_A(t) \rangle}}{\langle N_D \rangle \langle N_A \rangle} - 1 \quad (7)$$

$N_D(t)$ and $N_A(t)$ are the number of donor and acceptor photons in a bin at time $t$, $\langle \dots \rangle$ denotes an average in a given segment in a trajectory, and the upper bar indicates the average over segments that are longer than 20–95 ms. The results are shown in Supplementary Fig. 4.

## Data availability

Data supporting the findings of this manuscript are available from the corresponding author upon reasonable request.

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

## Acknowledgements

We thank W.A. Eaton, A. Szabo, I.V. Gopich, and R.B. Best for numerous helpful discussions and comments; P.G. Schultz for sharing the plasmid for the expression and incorporation of the unnatural amino acid; and J.M. Louis for advices and suggestions on protein expression and purification. This work was supported by the Intramural Research Program of the National Institute of Diabetes and Digestive and Kidney Diseases, NIH.

## Author contributions

J.-Y.K. and H.S.C. designed research and wrote the manuscript. J.-Y.K. performed research and data analysis. J.-Y.K. and F.M. developed protein expression and labeling protocols. J.Y developed data analysis tools. All authors discussed results and commented on the manuscript.

## Additional information

**Competing interests:** The authors declare no competing interests.

