## [Peer Review File · Nature Communications]

Reviewers' Comments:

Reviewer #1:

Remarks to the Author:

The manuscript presents a single molecule study of coupled folding-binding reaction between TAD and NCBD. The study focuses on the binding mechanism and concludes for an intermediate state of an encounter complex with a relatively long life time. Using a comprehensive analysis and modeling the authors concluded that the slow transition path time is related to the high energy intermediate. From the dependence of the kinetics on ionic strength it is concluded that electrostatic forces govern the intermediate states via the formation of non-native interactions. The presented results are novel and describe new insights on the mechanism of binding of IDPs. The connection to earlier studies and particularly the fly-casting mechanism is interesting. Comparing to the binding mechanism of Barnase-Barstar whose binding is uncoupled to the folding of the two individual proteins is very valuable and supports the main conclusion of this study that the encounter complex is governed by non-native electrostatic interactions.

Comments:

1. The measured rates can be affected by the setup of the experiments particularly, the positions of the fluorophores and if both are placed on the same protein and which protein is immobilized. How these parameters may affect the conclusions of this study?
2. How is this mechanism relevant in vivo given that the salt concentration in the cell is high and thus the stability of the encounter complex is expected to be low.

Reviewer #2:

Remarks to the Author:

This paper by Kim et al. describes a very interesting application of single molecule detection, to directly detect and quantify transient encounter complexes during the folding and binding of intrinsically disordered proteins (IDPs). In contrast with the traditional view of proteins, IDPs represent the significant fraction of the proteome that contains at least partially disordered sequences. An interesting aspect of IDPs is that many can undergo a coupled folding and binding process that is functionally relevant. A mechanism by which this process occurs is via induced folding, that is folding to the final structure occurs following binding. The encounter complex (referred to in the manuscript as the transient complex) involved in this process is hard to observe and is the subject of this study.

The authors use single molecule FRET (smFRET) to study the binding and folding of the p53 TAD domain and NCBD domain of CBP, a biologically interesting complex. Using immobilized dual-labeled TAD and added NCBD, in conjunction with an analysis of donor and acceptor photon streams, they carry out analysis to directly observe and quantify the occupation and lifetime of the TC. They find that the overall binding reaction is very fast, and that TC is relatively long lived (compared to previously measured transition path times for folding). A comparison with another electrostatically driven binding system of two folded proteins (barnase and barstar) shows a much shorter-lived TC. The authors also conclude that the longer lived TC is due to non-native electrostatic interactions based on a salt-dependence of the kinetics. While the problem and single molecule approach are interesting, and the data are of good quality, a few important concerns (listed below) make the manuscript as presented unsuitable for the readership of Nature Communications.

Importantly, the ideas of induced folding, encounter complexes and fly-casting are well discussed in the literature over many years. The authors presentation of the work seems to start with the assumption and imply that these ideas are surprising, which is confusing. As one example, in the introduction, they say "In IDP binding, the association is expected to be even slower because

folding should occur during the binding process and an IDP cannot bind as soon as it finds its target.”

The evidence for non-native electrostatic interactions (a central conclusion included in the title) in the TC seems weak. The authors only verified this point by screening electrostatics and a brief discussion of the bound structure. It is unclear how this is good evidence for the molecular origins of diffusion limited association, or the stabilization by non-native interactions. In a related note, it is unclear how non-native hydrophobic interactions can be ruled out.

It is also not clear how a rugged and flat energy landscape is ruled out as the cause of the relatively slow formation of the final bound structure.

The MLE method and its limitations for determining tTC and ETC should be better described in the text, including giving the reader a better idea for the possible parameter interdependencies and uncertainties (and errors). Presumably possible uncertainties are even higher for the 6-state model used. In a related note, it seems that a distribution of FRET efficiencies could be seen for an encounter complex, however this does not seem included in the analysis.

How did the authors determine the equilibrium dissociation constant used to design the experimental conditions? If not determined elsewhere for each NaCl concentration, I suggest authors determine them by appropriate methods, such as fluorescence anisotropy as implicated dissociation constants will be detectable by that simple ensemble method. Alternatively, intermolecular smFRET can also be used.

In terms of smFRET time-trajectories, it should be noted that data acquisition faster than 1ms (even 50 microseconds) has been reported, for example by the Munoz group. Such a time-resolution would improve direct visualization of the TC in the types of traces as in Figure 3. The authors should consider using such conditions for the experiment.

Reviewer #3:

Remarks to the Author:

This paper is an excellent study of IDP folding/binding dynamics on the single molecule level that allows the experimenter to directly observe the transition between unbound and bound/folded states. The transition time, while too fast to directly observe in binned trajectories (although it is obvious that it decreases significantly with increasing salt), can be observed by maximum likelihood analysis of all photon arrival times. The resulting times are quite long compared to the folding times observed in monomeric folding by this group. This leads to the conclusion is a high energy intermediate within the barrier between the unbound and bound states. Overall this is a significant result that will advance the field of IDP folding and binding, and I recommend publication after considering the following questions/suggestions:

- 1) I don't understand why the binned trajectories in Fig. 2a have a delay in the signal of both donor and acceptor. Was the laser turned on with a delay?
- 2) In fig. 3, part a shows photon trajectories binned to 100 microseconds and the log likelihood plots in part c. I assume the plots in a were not used for calculations in c since it should be using photon arrival times, but this should be made clearer.
- 3) I really don't think the fact that the association rate is diffusion limited without salt is that surprising since it is separable from the transition state time/rate. This just means that forming the transition state has a low barrier, but getting out of that state has a higher barrier. Shouldn't this mean the first peak in the red curve on Fig. 4 c should be lower? Perhaps increasing NaCl is only changing the depth of the unbound well, not the barrier height.

4) The stability of the transition state could be due to a decrease in free energy compared to the barriers on either side, as drawn in Fig. 4c, but I don't think the authors can rule out roughness in the landscape at the peak, since all they can measure is a time, not an equilibrium stability. This roughness would correspond to the diffusion coefficient in the prefactor using a Kramers formalism and there is no reason to expect D to be the same in the wells as at the barrier height.

Reviewer #4:

Remarks to the Author:

The manuscript by Kim describes an elegant set of experiments aimed at probing the association between an IDP (the TAD from p53) and the MG-like protein NCBD and, as a "control" also the association between barnase and barstar. In particular, the authors use state of the art smFRET experiments to probe the rapid timescales of association from the unbound to the bound state, the so-called transition path time. A careful analysis of the experiments reveals that the TAD/NCBD binding process is best described as a two-step process, involving first forming an intermediate (an encounter complex), which then rearranges into the final complex. As such, the experiments provide central and until now unknown information about the timescales involved in binding of IDPs. Further, by contrasting these observations with experiments on folded proteins, the authors are able to compare IDPs and folded proteins. The manuscript is well written, the experiments and analyses carefully performed and described, and the conclusions appear well justified. As such, I only have few comments on the work that the authors might consider.

1. It would be nice to place the results a bit better in the context of what has already been measured for this system. The bulk kinetic experiments of NCBD binding to a different IDP (ACTR) at low temperatures by Jemth has also been interpreted in terms of an on-pathway intermediate, but it is not clear whether this is the same and/or related. Also, in addition to the binding/unbinding kinetics, at least two other processes are known to occur in NCBD (major/minor state exchange and folding/unfolding), both occurring on the long-microsecond timescale. Presumably, by focusing only on direct binding events, these have no effect on the experiments here.

2. The authors find a strong correlation between fitted $\langle E_{TC} \rangle$ and the lifetime of the intermediate. Given that the assumption of $E_{TC} = 0.5(E_B + E_U)$ is not well justified, I suggest that the authors simply provide averages and errors from the joint analysis, or at least argue why the analysis using the assumed $\langle E_{TC} \rangle$ is better/relevant.

3. The authors find that for NCBD/TAD the lifetime of the TC is very long (0.6ms). Should one not be able to see effects of this in bulk experiments a burst phase, or missing amplitudes? Also, even using 100us binned data, should there not be a sizable number of events that clearly show the intermediate?

4. The authors show a rather strong dependency on the lifetime of the TC as well as the on rate on salt. Presumably this means that the two barriers surrounding the TC have rather different salt dependencies? In that case, could it be possible that there is a shift in rate-limiting step as a function of salt and/or could this affect $\langle E_{TC} \rangle$ [as also indicated in Fig. 4C]. Thus, could the observed correlation between fitted $\langle E_{TC} \rangle$ and the lifetime of the intermediate be more than a fitting artefact, and be providing information about a shift of the pathway?

5. In the model for analysis of the data (Fig. 3b) the two barriers for escaping the TC are the same (k_{TC}), but this is not the case in the final model (Fig. 4C). As far as I can see, there is no reason

why the two barriers should be the same, so presumably this is just for convenience in estimating the TC lifetime. What are the implications of this assumption?

6. The experiments on barnase/barstar are briefly compared to recent MD simulations, in particular stating that “when two molecules approach with different orientations, binding would not happen (Fig. 4d), which is consistent with the recent MD simulation result showing that the early intermediate states with different orientations would not affect overall binding kinetics significantly”. At the same time, the result of a very fast (<2 us) TP-time appears different from the slower rearrangements in the semi-bound configurations observed in simulations. It would be useful to interpret the results in the context of those simulations to see whether they agree. In particular, since most MD force fields are known to be overly sticky it would be relevant to know whether the slow reorganization of the TC in simulations is realistic or not.

Reviewer #1 (Remarks to the Author):

The manuscript presents a single molecule study of coupled folding-binding reaction between TAD and NCBD. The study focuses on the binding mechanism and concludes for an intermediate state of an encounter complex with a relatively long life time. Using a comprehensive analysis and modeling the authors concluded that the slow transition path time is related to the high energy intermediate. From the dependence of the kinetics on ionic strength it is concluded that electrostatic forces govern the intermediate states via the formation of non-native interactions. The presented results are novel and describe new insights on the mechanism of binding of IDPs. The connection to earlier studies and particularly the fly-casting mechanism is interesting.

Comparing to the binding mechanism of Barnase-Barstar whose binding is uncoupled to the folding of the two individual proteins is very valuable and supports the main conclusion of this study that the encounter complex is governed by non-native electrostatic interactions.

→ We thank Reviewer #1 very much for considering our manuscript highly. We revised the manuscript according to reviewer's comments below.

Comments:

1. The measured rates can be affected by the setup of the experiments particularly, the positions of the fluorophores and if both are placed on the same protein and which protein is immobilized. How these parameters may affect the conclusions of this study?

→ Reviewer's question is reasonable because perturbation by attaching relatively large fluorophores and immobilization can be always issues in single-molecule FRET experiments. We added additional data in Supplementary Fig. 2 that shows the FRET efficiency distributions measured from freely-diffusing molecules are very similar to those from immobilized molecules over a wide range of NaCl concentration. A peak at $E \sim 0.1$ in the free diffusion data results from molecules without an active acceptor (donor-only) that cannot be separated. The similarity in histograms from the two different experiments indicates that the immobilization effect on the binding dynamics is very small. In addition to Supplementary Fig. 2, we added a brief description of the free diffusion experiment in Methods section (page 12).

“In the free diffusion experiment, 80 pM of TAD was mixed with NCBD at various NaCl concentrations. The same chemicals used for the immobilization experiment were added to reduce dye photobleaching and blinking. To prevent protein sticking to a glass surface, 0.01% Tween 20 was used. The solution was illuminated at 22 μ W and fluorescence bursts were measured 10 μ m above the glass surface.”

We also think the influence by the fluorophore attachment is very small. From our previous experience, the effects of immobilization and fluorophores such as Alexa 488, 594, and 647, two of which we used in this study, on the kinetics of protein folding have been small, only a factor of two, compared to ensemble measurements. (Chung et al. *PNAS*, vol 106, 11837 (2009) and Chung et al. *Chem Phys*, vol 422, 229 (2013)) To minimize the effect of fluorophores further, we attached these to the two ends of TAD. Since several residues at the N- and C-termini are disordered in addition to the flexible linkers attached to the dyes, we expect the perturbation of the binding and folding dynamics by dyes will be smaller than that in the previous works above.

2. How is this mechanism relevant *in vivo* given that the salt concentration in the cell is high and thus the stability of the encounter complex is expected to be low.

→ In this work, we tried to measure the lifetime of the transient complex (TC) up to 150 mM NaCl, which is a physiologically relevant salt concentration as the reviewer pointed out. However, it was not possible to determine at this concentration because the kinetics becomes very fast and the FRET efficiency difference between the bound and unbound states are too close. However, the lifetime of TC is ~ 50 μ s at 90 mM NaCl, which is not so different from 150 mM, and the association rate at 150 mM is only 2/3 of that at 90 mM. From this trend of the data, we expect the lifetime of TC at 150 mM is also quite long (tens of μ s) and not short like that of barnase and barstar. Therefore, we think the mechanism is relevant to *in vivo* condition as well.

Reviewer #2 (Remarks to the Author):

This paper by Kim et al. describes a very interesting application of single molecule detection, to directly detect and quantify transient encounter complexes during the folding and binding of intrinsically disordered proteins (IDPs). In contrast with the traditional view of proteins, IDPs represent the significant fraction of the proteome that contains at least partially disordered sequences. An interesting aspect of IDPs is that many can undergo a coupled folding and binding process that is functionally relevant. A mechanism by which this process occurs is via induced folding, that is folding to the final structure occurs following binding. The encounter complex (referred to in the manuscript as the transient complex) involved in this process is hard to observe and is the subject of this study.

The authors use single molecule FRET (smFRET) to study the binding and folding of the p53 TAD domain and NCBD domain of CBP, a biologically interesting complex. Using immobilized dual-labeled TAD and added NCBD, in conjunction with an analysis of donor and acceptor photon streams, they carry out analysis to directly observe and quantify the occupation and lifetime of the TC. They find that the overall binding reaction is very fast, and that TC is relatively long lived (compared to previously measured transition path times for folding). A comparison with another electrostatically driven binding system of two folded proteins (barnase and barstar) shows a much shorter-lived TC. The authors also conclude that the longer lived TC is due to non-native electrostatic interactions based on a salt-dependence of the kinetics. While the problem and single molecule approach are interesting, and the data are of good quality, a few important concerns (listed below) make the manuscript as presented unsuitable for the readership of Nature Communications.

→ We are very grateful that Reviewer #2 considers that the problem and our experimental approach are interesting and our work is of good quality. We also thank Reviewer #2 very much for careful reading and giving us valuable comments. We revised manuscript according to reviewer's comments and concerns as listed below and hope that now it is suitable for publication in *Nature Communications*.

1. Importantly, the ideas of induced folding, encounter complexes and fly-casting are well discussed in the literature over many years. The authors presentation of the work seems to start with the assumption and imply that these ideas are surprising, which is confusing. As one example, in the introduction, they say "In IDP binding, the association is expected to be even slower because folding should occur during the binding process and an IDP cannot bind as soon as it finds its target."

→ The reviewer is correct that those ideas have been discussed for many years to explain binding of IDPs. However, how much time is needed for folding of an IDP and how long the encounter complex (or transient complex) would last have not been measured as also pointed out by Review #4. The important discovery in our study is that IDP binding can be so fast (diffusion-limited) and this is possible not solely because of the effects expected from the above ideas but also because the transient complex (TC) is so stable and lasts long, which we wanted to emphasize in the manuscript. In addition, this observation is possible only by direct measurement of the lifetime of TC, which we believe is an important advance for understanding the mechanism of IDP binding.

Although we have not intended to give an impression that the ideas the reviewer mentioned are surprising, after careful reading, we found several sentences that may read as the reviewer pointed out. Therefore, we rephrased those sentences as listed below and added references for encounter complexes.

(page 2) “In IDP binding, the association is expected to be even slower because folding should occur during the binding process and an IDP **may dissociate easily before folding even if it encounters a target protein.**”

(page 3) “We define a transient complex (TC, **also known as encounter complex**^{27–30}) as a representative state appearing in the collection of binding pathways that lead unbound (disordered) TAD to a bound complex. We aim to describe the mechanism of diffusion-limited association of TAD and NCBD in terms of the properties of the transient complex. So far, the average picture of intermediate species during binding **including encounter complexes** has been characterized by NMR spectroscopy³¹ such as relaxation dispersion³² and paramagnetic relaxation enhancement^{28,33}.”

2. The evidence for non-native electrostatic interactions (a central conclusion included in the title) in the TC seems weak. The authors only verified this point by screening electrostatics and a brief discussion of the bound structure. It is unclear how this is good evidence for the molecular origins of diffusion limited association, or the stabilization by non-native interactions. In a related note, it is unclear how non-native hydrophobic interactions can be ruled out.

→ If the ionic strength dependence is marginal, we would agree with reviewer’s point that the evidence is weak. However, we cannot think of any origin other than the involvement of strong electrostatic interactions to see very large ionic strength dependences of both association rate and lifetime of TC. Since the native interaction (from the structure) is hydrophobic, we believe it is reasonable to conclude that these electrostatic interactions must be non-native. More evidence with molecular details may be provided only by MD simulations when the results are consistent with the experimental observations. Nevertheless, we agree that our experiments cannot exclude non-native hydrophobic interactions. Therefore, we included this point in the text on page 9,

“Therefore, the electrostatic interactions must be involved in the formation of the transient complex and these interactions are non-native, **although non-native hydrophobic interactions can also contribute**^{32,48}.”

3. It is also not clear how a rugged and flat energy landscape is ruled out as the cause of the relatively slow formation of the final bound structure.

→ We didn't rule out a rugged/flat energy landscape. Actually, non-native electrostatic interactions can slow folding of TAD and the formation of the bound complex. This point is discussed in the second last paragraph of Discussion on page 10,

“In the case of diffusion-limited association at 0 mM NaCl, the lifetime of TC (several hundred μ s) corresponds to the folding time of TAD ($t_{TC} = 1/[k_f + k_-]$ and $k_- \ll k_f$ in Fig. 4c). This time is actually much longer than folding times of many single domain (α -helical) fast-folding proteins that can fold in several μ s or even shorter⁵⁶. The slow folding rate of TAD may result from the increased internal friction by non-native interactions as found for folding of a designed protein, α_3D ⁴³. Between the two opposite effects of non-native interactions on the overall rate of diffusion-limited association ($k_A = k_+k_f/[k_f + k_-]$), the enhancement by avoiding dissociation (reduced k_-)²² is much larger than the reduction due to slower folding by a few fold (reduced k_f)⁴³.”

4. The MLE method and its limitations for determining t_{TC} and E_{TC} should be better described in the text, including giving the reader a better idea for the possible parameter interdependencies and uncertainties (and errors). Presumably possible uncertainties are even higher for the 6-state model used. In a related note, it seems that a distribution of FRET efficiencies could be seen for an encounter complex, however this does not seem included in the analysis.

→ The errors of the parameters in the determination of t_{TC} and E_{TC} (6-state model) are listed in Supplementary Table 2. Error bars of t_{TC} are also shown in Fig. 3E. It is somewhat unclear to us what to discuss about the limitations of the likelihood method. Because errors of t_{TC} is relatively small, 10 – 15% of the corresponding values due to the high likelihood values, we don't think this would affect any of the conclusions.

The reviewer is correct that there would be a distribution of the FRET efficiency of the transient complex (E_{TC}) because there would be various binding pathways as discussed at the end of the manuscript (page 10). Inspired by the comment, we tried the simplest model for the analysis of the heterogeneity of binding pathways: two different pathways (i.e. two different E_{TC} and t_{TC}). However, fitting did not converge. It may be because the current data is not sufficiently good for this analysis or the FRET efficiencies of TC in different pathways may be similar because TAD in TC is disordered as we discussed in the manuscript. This would be an interesting topic for us to investigate in the future. Thank you!

5. How did the authors determine the equilibrium dissociation constant used to design the experimental conditions? If not determined elsewhere for each NaCl concentration, I suggest authors determine them by appropriate methods, such as fluorescence anisotropy as implicated dissociation constants will be detectable by that simple ensemble method. Alternatively, intermolecular smFRET can also be used.

→ We performed free diffusion experiments first and determined an appropriate concentration of the binding partner where bound and unbound populations are comparable at each NaCl concentration. Then, we carried out immobilization experiment at the same concentration of NCBP. The accurate dissociation constant can be obtained from the maximum likelihood analysis of the immobilization data as listed in Supplementary Table 1. (We also added the free diffusion data in Supplementary Fig. 2 to answer Comment #1 of Reviewer #1.)

6. In terms of smFRET time-trajectories, it should be noted that data acquisition faster than 1ms (even 50 microseconds) has been reported, for example by the Munoz group. Such a time-resolution would improve direct visualization of the TC in the types of traces as in Figure 3. The authors should consider using such conditions for the experiment.

→ The experimental condition for the data presented in Fig. 3 is very similar to the condition that the reviewer mentioned. The bin time is 200 μ s for the trajectories in Fig. 3a, which show gradual changes in the FRET efficiency as indicated in the figure legend (at other salt concentrations, bin time is 100 μ s). However, gradual FRET efficiency changes during binding and dissociation is not always clearly observed, and maximum likelihood analysis of photon trajectories is still necessary for the determination of the accurate lifetimes of TC.

We revised the text to clarify this point and cited the work by the Munoz group as suggested on page 5,

“The lifetime of TC should be shorter than the bin time of 1 ms because the transitions appear instantaneous in Fig. 2a. For a better time resolution, we raised the laser illumination intensity by a factor of 5 - 10 and reduced the bin time to 100 - 200 μ s. (Fig. 3a)^{34,40,41}. As indicated by yellow shades, there are several bins (200 μ s bin time) with intermediate FRET efficiencies between the bound ($E \sim 0.6$) and unbound ($E \sim 0.25$) states, suggesting that an additional state exists between the bound and unbound states. Since this state with an intermediate FRET efficiency is not always obviously detectable in a binned trajectory, we performed a maximum likelihood analysis of photon trajectories ...”

Reviewer #3 (Remarks to the Author):

This paper is an excellent study of IDP folding/binding dynamics on the single molecule level that allows the experimenter to directly observe the transition between unbound and bound/folded states. The transition time, while too fast to directly observe in binned trajectories (although it is obvious that it decreases significantly with increasing salt), can be observed by maximum likelihood analysis of all photon arrival times. The resulting times are quite long compared to the folding times observed in monomeric folding by this group. This leads to the conclusion is a high energy intermediate within the barrier between the unbound and bound states. Overall this is a significant result that will advance the field of IDP folding and binding, and I recommend publication after considering the following questions/suggestions:

→ We thank Reviewer #3 very much for considering our manuscript very highly. We revised the manuscript according to reviewer’s comments below.

1) I don't understand why the binned trajectories in Fig. 2a have a delay in the signal of both donor and acceptor. Was the laser turned on with a delay?

→ Yes, it is correct. At this (moderately) high illumination intensity, photobleaching is very fast. Therefore, we started data collection before turning on the laser not to lose the beginning part of trajectories.

2) In fig. 3, part a shows photon trajectories binned to 100 microseconds and the log likelihood plots in part c. I assume the plots in a were not used for calculations in c since it should be using photon arrival times, but this should be made clearer.

→ The reviewer is correct. Fig. 3a is just for the visualization of gradual FRET efficiency changes for selected trajectories. We made this point clearer by adding a sentence in the caption of Fig. 3a. By the way, there was a mistake for the bin time. It is 200 μ s instead of 100 μ s.

“The accurate lifetime of TC was determined using the maximum likelihood analysis of photon trajectories without binning (see Fig. 3e).”

3) I really don't think the fact that the association rate is diffusion limited without salt is that surprising since it is separable from the transition state time/rate. This just means that forming the transition state has a low barrier, but getting out of that state has a higher barrier. Shouldn't this mean the first peak in the red curve on Fig. 4 c should be lower? Perhaps increasing NaCl is only changing the depth of the unbound well, not the barrier height.

→ Although we presented Fig. 4c to explain the mechanism, we note that this can be still confusing. Unlike protein folding, which is a uni-molecular reaction, binding is a bi-molecular reaction and the relative stability of the unbound state to the bound state as well as TC depend on the concentration of the binding partner. The apparent association rate is also proportional to the concentration of the binding partner ($k_{A,app} = k_A \times [NCBD]$). Therefore, if [NCBD] is very large, the unbound state is largely destabilized and the reaction will be downhill from unbound state to TC in Fig. 4c. However, this has nothing to do with diffusion-limited reaction because it is determined by k_A not $k_{A,app}$. The free energy diagram in Fig. 4c is for the case with the same arbitrary NCBD concentration with different [NaCl], where apparent association and dissociation rates are comparable. To avoid confusion, we revised the caption of Fig. 4c as,

“Kinetic scheme of the association via the formation of TC, and proposed free energy surface of TAD/NCBD binding with an arbitrary NCBD concentration at different NaCl concentrations.”

In addition, the modulation of the relative heights of the two barriers from TC to the unbound and bound states is necessary to explain the diffusion-limited association and its strong ionic strength dependence. For the diffusion-limited association, molecules in TC should form a bound complex rather than dissociates. This means that the barrier on the left side in Fig. 4c should be higher than the barrier on the right side (k_r is smaller than k_f). At higher ionic strength, the relative heights of the two barriers will switch because molecules preferentially dissociate from TC rather than form the bound complex. Although this point is explained on page 9, we describe this in the caption of Fig. 4c as

“As [NaCl] is increased, the formation of TC becomes slower (reduced k_r) due to the increased charge screening effect, and TC becomes less stable and dissociates more easily (increased k_r , lowered dissociation barrier) before TAD folds to form a fully bound complex. Both effects decrease k_A .”

4) The stability of the transition state could be due to a decrease in free energy compared to the barriers on either side, as drawn in Fig. 4c, but I don't think the authors can rule out roughness in the landscape

at the peak, since all they can measure is a time, not a equilibrium stability. This roughness would correspond to the diffusion coefficient in the prefactor using a Kramers formalism and there is no reason to expect D to be the same in the wells as at the barrier height.

→ The reviewer's point is absolutely correct that the change in roughness of the landscape (diffusion coefficient) can affect the result. Unfortunately, we cannot answer this question definitively because separating the effects of stability and diffusion coefficient on the rate is extremely difficult in this case. However, it is hard to imagine that more than 10 fold difference in the lifetime of TC between 0 and 150 mM NaCl solely results from the change in the diffusion coefficient as electrostatic interactions is not completely eliminated at 150 mM NaCl. (Factor of 10 changes in the diffusion coefficient was observed for α_3D folding when the non-native salt bridges are completely eliminated. Chung et al. Science v349, 1504, (2015)) Large changes in the relative stability of bound and unbound states and association and dissociation rates also suggest there would be modulation in the relative heights of the two barriers of TC in Fig. 4c. Nevertheless, there should be a contribution of the roughness to the long lifetime of TC and we added this point in the text (page 9) as

“Non-native interactions at low ionic strength also suggest slow diffusion along the reaction coordinate, which contributes to the increased t_{TC} along with the stability of TC similar to the increased folding transition path time of α_3D by non-native salt-bridge formation⁴³.”

In addition, the effect of the roughness on the mechanism can also be found in the discussion on page 10.

“It is also worth noting that the enhancement of association by non-native interactions is an important mechanism for IDP binding, whereas in protein folding, non-native interactions do not affect the folding mechanism⁵⁴, but slow the folding process^{43,55}. In the case of diffusion-limited association at 0 mM NaCl, the lifetime of TC (several hundred μ s) corresponds to the folding time of TAD ($t_{TC} = 1/[k_f + k_-]$ and $k_- \ll k_f$ in Fig. 4c). This time is actually much longer than folding times of many single domain (α -helical) fast-folding proteins that can fold in several μ s or even shorter⁵⁶. The slow folding rate of TAD may result from the increased internal friction by non-native interactions as found for folding of a designed protein, α_3D ⁴³. Between the two opposite effects of non-native interactions on the overall rate of diffusion-limited association ($k_A = k_+k_f/[k_f + k_-]$), the enhancement by avoiding dissociation (reduced k_-)²² is much larger than the reduction due to slower folding by a few fold (reduced k_f)⁴³.”

Reviewer #4 (Remarks to the Author):

The manuscript by Kim describes an elegant set of experiments aimed at probing the association between an IDP (the TAD from p53) and the MG-like protein NCBD and, as a “control” also the association between barnase and barstar. In particular, the authors use state of the art smFRET experiments to probe the rapid timescales of association from the unbound to the bound state, the so-called transition path time. A careful analysis of the experiments reveals that the TAD/NCBD binding process is best described as a two-step process, involving first forming an intermediate (an encounter complex), which then rearranges into the final complex. As such, the experiments provide central and until now unknown information about the timescales involved in binding of IDPs. Further, by

contrasting these observations with experiments on folded proteins, the authors are able to compare IDPs and folded proteins. The manuscript is well written, the experiments and analyses carefully performed and described, and the conclusions appear well justified. As such, I only have few comments on the work that the authors might consider.

→ We are very grateful that Reviewer #4 considers our manuscript very highly. We revised the manuscript according to reviewer's comments below.

1. It would be nice to place the results a bit better in the context of what has already been measured for this system. The bulk kinetic experiments of NCBD binding to a different IDP (ACTR) at low temperatures by Jemth has also been interpreted in terms of an on-pathway intermediate, but it is not clear whether this is the same and/or related. Also, in addition to the binding/unbinding kinetics, at least two other processes are known to occur in NCBD (major/minor state exchange and folding/unfolding), both occurring on the long-microsecond timescale. Presumably, by focusing only on direct binding events, these have no effect on the experiments here.

→ It is true that the kinetic data of ACTR and NCBD binding in the previous work was fitted to a model with intermediate states (Dogan et al. JBC, 2012). However, the lifetime of the intermediate state is ~ 15 - 20 ms, which is much longer than the lifetime of TC in our study. So, as the reviewer mentioned, these may not be related. Since it is a different binding system, we cited this work in Introduction instead of discussing it in detail. In addition, NCBD is known to be flexible by itself and the bound conformation varies upon binding partners. However, as pointed out by the reviewer, we don't expect any effect of the NCBD dynamics on binding to TAD because we are analyzing direct binding events. We added a sentence mentioning the flexibility and binding of NCBD to different binding partners with appropriate references in Introduction (page 2).

“NCBD is also a flexible protein like a molten-globule^{17,18} and interacts with multiple binding partners¹⁹ including TAD and the activation domain of SRC-3 (ACTR)^{20,21}.”

2. The authors find a strong correlation between fitted $\langle E_{TC} \rangle$ and the lifetime of the intermediate. Given that the assumption of $E_{TC}=0.5(E_B+E_U)$ is not well justified, I suggest that the authors simply provide averages and errors from the joint analysis, or at least argue why the analysis using the assumed $\langle E_{TC} \rangle$ is better/relevant.

→ As the reviewer pointed out, the joint analysis would give a more accurate result. One reason that we also plotted likelihoods with $E_{TC} = (E_B + E_U)/2$ is that it provides more rigorous comparison between the instantaneous transition model ($t_{TC} = 0$) and the model with a finite lifetime of TC. How reliable the determined t_{TC} is can be visualized by the peak of the likelihood compared to the 95% confidence level. In this plot, the likelihood of the two models can be directly compared because the analysis is designed such that the number of parameters is the same (only t_{TC} is changed). In addition, in this way the upper bound of TC lifetime can also be determined when the peak is lower than the confidence level (150 mM NaCl). On the other hand, in the joint analysis, the number of parameters of the three (or six) state model with the finite lifetime of TC is more than that of the instantaneous transition model, and an appropriate statistical comparison is required such as BIC, which usually gives a preference to a model with more parameters. Since the presence of TC with a long lifetime is assured by a significantly high likelihood peak in the analysis with $E_{TC} = (E_B + E_U)/2$, we can safely trust

the joint analysis result as well. Therefore, we prefer to present both analysis, but if it is necessary we can move the analysis with $E_{TC} = (E_B + E_U)/2$ to Supplementary Material.

3. The authors find that for NCBD/TAD the lifetime of the TC is very long (0.6ms). Should one not be able to see effects of this in bulk experiments a burst phase, or missing amplitudes? Also, even using 100us binned data, should there not be a sizable number of events that clearly show the intermediate?

→ TC in our study may not be a well-defined state although we analyzed with the three-state model, and it appears only when binding occurs like the transition path. Since binding transitions are not synchronized (stochastic), TC won't be detectable in bulk experiments such as stopped flow or temperature jump experiments unless there is a fair amount of population change in TC due to a very low barrier height (as in the WW domain folding experiment by the Gruebele group, Nature 2003), which is not the case for TAD and NCBD binding.

At the lowest salt concentration (0 mM NaCl), the average FRET efficiency of TC is quite low (0.34), which differ from that of the unbound state (0.26) only by 0.08. Therefore, TC is not so easily distinguishable in binned trajectories. Nevertheless, we looked into the data and added 2 more transitions in Fig. 3a, in which the lifetime of TC looks sufficiently long. In addition, we apologize that we made a mistake in Fig. 3a. This data (0 mM NaCl) was collected at the illumination intensity 5 times higher than that for the kinetics measurement whereas it was 10 - 20 times higher at all other conditions. Therefore, the bin time is 200 μ s in Fig. 3a instead of 100 μ s, and the lifetime of TC in the presented trajectories would be longer than 400 μ s.

4. The authors show a rather strong dependency on the lifetime of the TC as well as the on rate on salt. Presumably this means that the two barriers surrounding the TC have rather different salt dependencies? In that case, could it be possible that there is a shift in rate-limiting step as a function of salt and/or could this affect $\langle E_{TC} \rangle$ [as also indicated in Fig. 4C]. Thus, could the observed correlation between fitted $\langle E_{TC} \rangle$ and the lifetime of the intermediate be more than a fitting artefact, and be providing information about a shift of the pathway?

→ The review's point that the relative height of the two barriers change with the salt concentration is consistent with our interpretation as drawn in Fig. 4c. However, we don't think this would cause a shift of the rate-limiting step because the major barrier is always located between unbound and TC. A barrier from TC to the bound state is only a small addition, which cannot be the rate-limiting step. Our interpretation for the observation that the FRET efficiency of TC (E_{TC}) increases with the increasing NaCl concentration is that TC is still largely disordered and becomes more compact at a higher NaCl concentration as seen for the unbound state. (Of course, we don't think this is a fitting artifact.)

This is mentioned on page 5,

“ E_{TC} increases with the increasing NaCl concentration similar to E_U (Fig. 3e inset), indicating unstructured TAD in TC also becomes more compact at higher ionic strength.”

5. In the model for analysis of the data (Fig. 3b) the two barriers for escaping the TC are the same (k_{TC}), but this is not the case in the final model (Fig. 4C). As far as I can see, there is no reason why

the two barriers should be the same, so presumably this is just for convenience in estimating the TC lifetime. What are the implications of this assumption?

→ The review's point is correct. The two barriers at TC are not necessarily the same, but we made the two escaping rates the same for the convenience in the analysis that determines only the lifetime of TC rather than the relative flux toward the bound and unbound states. We cannot determine the relative flux because we are analyzing only the transition part of the trajectories (non-equilibrium). However, even if we make the two escaping rates arbitrarily different (e.g., 1:2 ratio), the lifetime of TC is the inverse of the sum of these two rates and the determined lifetime value is the same regardless of this ratio. Therefore, this assumption does not affect the analysis results.

6. The experiments on barnase/barstar are briefly compared to recent MD simulations, in particular stating that “when two molecules approach with different orientations, binding would not happen (Fig. 4d), which is consistent with the recent MD simulation result showing that the early intermediate states with different orientations would not affect overall binding kinetics significantly”. At the same time, the result of a very fast (<2 us) TP-time appears different from the slower rearrangements in the semi-bound configurations observed in simulations. It would be useful to interpret the results in the context of those simulations to see whether they agree. In particular, since most MD force fields are known to be overly sticky it would be relevant to know whether the slow reorganization of the TC in simulations is realistic or not.

→ We expanded the discussion of this work (see below and page 8) in addition to Supplementary Fig. 8 that shows the analysis does not depend on E_{TC} in the model. This result may not be consistent with the simulation result. As the reviewer pointed out, the hydrophobic force field may induce additional reconfiguration time in TC in simulation. However, it seems that the detailed comparison should be made by additional simulations with different/improved force fields.

“In the simulation, the lifetime of the transition state is 2 μ s, which is close to the upper bound of the lifetime of TC. Although the lifetime of late intermediate states is $\sim 10 \mu$ s, which is not detected as a significant peak in the likelihood plots with different E_{TC} values in our analysis (Supplementary Fig. 8), the overall duration of binding in the simulation is still much shorter than that of TAD and NCBD. The native-like transition state and other transient intermediate states explain why the lifetime of TC of barnase and barstar binding is very short compared to that of TAD and NCBD.”

Trivial:

1. p. 13: Fog -> For

→ Fixed. Thank you.

Reviewers' Comments:

Reviewer #1:

Remarks to the Author:

The authors have addressed satisfactorily my original comments

Reviewer #2:

Remarks to the Author:

The authors have done a good job of responding to my previous round of review comments. The work addresses and provides interesting new insight into an important problem in the biophysics of intrinsically disordered proteins. The revised manuscript is improved and will be of substantial interest to the readers of Nature Communications.

Reviewer #3:

Remarks to the Author:

The authors have addressed my questions and I recommend publication.

Reviewer #4:

Remarks to the Author:

The authors have done a nice job of responding to the reviews. My only remaining comment is the answer to Reviewer #4 (me), point 5 relating to the relative height of the two barriers. I fully understand the reason for and effect of this choice. I still would suggest to the authors to mention that this is a choice for convenience of analysis, and not something seen in the data or expected on physical grounds. Yes, the two barriers could not be *extremely* different, but they could easily be quite different. These plots and models affect how people think about dynamic processes, and hence it is important to state clearly what are assumptions and what are results. Apart from this, this is (still) a very nice paper.

Reviewer #1 (Remarks to the Author):

The authors have addressed satisfactorily my original comments

→ Thank you very much!

Reviewer #2 (Remarks to the Author):

The authors have done a good job of responding to my previous round of review comments. The work addresses and provides interesting new insight into an important problem in the biophysics of intrinsically disordered proteins. The revised manuscript is improved and will be of substantial interest to the readers of Nature Communications.

→ Thank you very much!

Reviewer #3 (Remarks to the Author):

The authors have addressed my questions and I recommend publication.

→ Thank you very much!

Reviewer #4 (Remarks to the Author):

*The authors have done a nice job of responding to the reviews. My only remaining comment is the answer to Reviewer #4 (me), point 5 relating to the relative height of the two barriers. I fully understand the reason for and effect of this choice. I still would suggest to the authors to mention that this is a choice for convenience of analysis, and not something seen in the data or expected on physical grounds. Yes, the two barriers could not be *extremely* different, but they could easily be quite different. These plots and models affect how people think about dynamic processes, and hence it is important to state clearly what are assumptions and what are results. Apart from this, this is (still) a very nice paper.*

→ Thank you very much! We added the following sentence in the figure 3 legend for the clarification.

“The two rate coefficients of the transitions from TC to the bound and unbound states are set to be equal (k_{TC}) for the convenience of the analysis, which does not reflect the actual relative heights of the two barriers in Fig. 4c.”